# Status of the Current Treatment Options and Potential Future Targets in Uterine Leiomyosarcoma: A Review

**DOI:** 10.3390/cancers14051180

**Published:** 2022-02-24

**Authors:** Hiroshi Asano, Toshiyuki Isoe, Yoichi M. Ito, Naoki Nishimoto, Yudai Watanabe, Saki Yokoshiki, Hidemichi Watari

**Affiliations:** 1Department of Obstetrics and Gynecology, Hokkaido University Graduate School of Medicine, Sapporo 060-8638, Japan; asano.hj@pop.med.hokudai.ac.jp; 2Clinical Research and Medical Innovation Center, Promotion Unit, Institute of Health Science Innovation for Medical Care, Hokkaido University Hospital, Sapporo 060-8648, Japan; toshiyuki.isoe@med.hokudai.ac.jp (T.I.); watanabe-yudai@pop.med.hokudai.ac.jp (Y.W.); yokoshiki@med.hokudai.ac.jp (S.Y.); 3Data Science Center, Promotion Unit, Institute of Health Science Innovation for Medical Care, Hokkaido University Hospital, Sapporo 060-8648, Japan; ito-ym@med.hokudai.ac.jp (Y.M.I.); naoki.nishimoto@pop.med.hokudai.ac.jp (N.N.)

**Keywords:** uterine leiomyosarcoma, next-generation sequencing, genomic, molecular-targeted drugs, HRD, *BRCA*, PARP inhibitors

## Abstract

**Simple Summary:**

Uterine leiomyosarcoma (uLMS) is a rare and aggressive mesenchymal malignancy. Although approximately 65% of patients are diagnosed in stage I, more than 50% have relapsed disease, and effective therapies for recurrent and advanced cases are limited. This review summarizes the current standard therapies for uLMS and the molecular properties of uLMS and describes the status of promising novel molecular-targeted therapies.

**Abstract:**

Uterine leiomyosarcoma (uLMS) is the most common subtype of mesenchymal tumors in the uterus. This review aims to summarize the current standard therapies and the molecular properties of uLMS for novel molecular-targeted therapies. Although 65% of uLMS cases are diagnosed in stage I, the 5-year overall survival rate is less than 60%. The only effective treatment for uLMS is complete and early resection, and chemotherapy is the main treatment for unresectable advanced or recurrent cases. No chemotherapy regimen has surpassed doxorubicin monotherapy as the first-line chemotherapy for unresectable advanced or recurrent cases in terms of overall survival in phase 3 trials. As a second-line treatment, pazopanib, trabectedin, and eribulin are used, but their therapeutic effects are not sufficient, highlighting the urgent need for development of novel treatments. Recent developments in gene analysis have revealed that homologous recombination deficiency (HRD), including breast cancer susceptibility gene 2 (*BRCA2*) mutations, are frequently observed in uLMS. In preclinical studies and several case series, poly(adenosine diphosphate-ribose)polymerase inhibitors showed antitumor effects on uLMS cell lines with *BRCA2* mutations or HRD and in recurrent or persistent cases of uLMS with *BRCA2* mutations. Thus, HRD, including *BRCA* mutations, may be the most promising therapeutic target for uLMS.

## 1. Introduction

Soft tissue sarcoma (STS) is a rare malignancy that originates in soft tissues derived from the mesoderm, such as fibrous tissue, adipose tissue, muscle tissue, and vascular tissue. It can occur anywhere in the body. The incidence of STS is less than 1% of all malignancies [1,2]. Data from the American Cancer Society show that 13,460 cases of STS and 5350 deaths due to STS occur annually in the United States [2]. The number of cases is considerably lower in Japan, with 1769 cases registered in 2015 by the Soft Tissue Tumor Registry in Japan and by The Japanese Orthopaedic Association Committee on Musculoskeletal Tumors [3,4]. Because STS originates from various types of normal tissues, histologic subtypes are highly diverse and are classified into more than 50 types. Leiomyosarcoma (LMS) is one of the most prevalent subtypes of STS, accounting for 6% to 10% of the cases [1]. LMS is classified into uterine LMS (uLMS), which originates in the uterus, and nonuterine LMS (nuLMS), which originates in other soft tissues.

Sarcomas in the uterus are classified into carcinosarcoma and mesenchymal tumors, and uLMS is the most common subtype of mesenchymal tumors in the uterus. The histopathologic diagnosis of uLMS widely applies the diagnostic criteria proposed by Hendrickson and Kempson, which comprehensively evaluate cellular atypia, mitotic index, and coagulative necrosis [5]. Although uLMS is the most common subtype of uterine sarcoma, only approximately 200 new cases are reported annually in Japan [6]. Owing to its rarity, a novel treatment strategy has not been developed. This review will summarize the status of uLMS and the possibilities of novel treatments based on recent molecular analyses.

## 2. Literature Search and Selection

We searched articles published up to November 2021 in the PubMed database (https://pubmed.ncbi.nlm.nih.gov/ accessed on 10 November 2021) using the following keywords: “sarcoma and phase 3” or “uterine leiomyosarcoma and clinical trial”. We checked all titles and abstracts of the searched articles and confirmed the clinical trials for STS, including uLMS. We reviewed the results of the clinical trials by obtaining the full text of phase 2 and 3 trials. We included articles using the following criteria: phase 2 or 3 clinical trials of chemotherapy for advanced or recurrent STS; phase 3 trials for STS including LMS; phase 2 trials for uLMS alone published after 2000 (Figure 1). We excluded all papers without accessible full text. We searched the reference list for selected studies to identify additional relevant articles. As a result, we found 34 articles that reported phase 3 clinical trials about the efficacy of antitumor or molecular-targeted drugs on recurrent and/or persistent STS, including uLMS; 17 trials were first-line chemotherapy, and the remaining seven trials were second-line chemotherapy or beyond. Furthermore, we reviewed clinical trials targeting uLMS alone, including 21 phase 2 clinical trials. Then, we searched for research on molecular features of uLMS using the keyword “leiomyosarcoma and genomic” via PubMed as well as for ongoing clinical trials on uLMS in ClinicalTrials.gov (https://clinicaltrials.gov/ accessed on 10 November 2021) using the keywords “uterine leiomyosarcoma”, “sarcoma”, and “PARP inhibitor”.

## 3. Standard Therapy for uLMS

### 3.1. uLMS Staging and the Overview of Prognosis

uLMS is staged differently from endometrial cancer. In principle, staging is determined based on the International Federation of Gynecology and Obstetrics (FIGO) 2008 classification after evaluating the degree of tumor progression according to laparotomy findings. According to the tumor registry of the Japan Society of Obstetrics and Gynecology, the distribution of uLMS cases in Japan is 65% for stage I disease, 10% for stage II, 8% for stage III, and 17% for stage IV [6]. Furthermore, the Japanese Gynecologic Oncology Group (JGOG) 2049s analyzed the treatment and prognosis of uLMS in Japan and found that it has a poor prognosis, with 5-year overall survival rates (5-year OS) of 57.1%, 44.6%, 20.4%, and 23.2% for stage I, II, III, and IV disease, respectively [7]. Table 1 summarizes the staging (FIGO 2008 classification), distribution, and prognosis of uLMS in Japan.

### 3.2. Surgical Procedures and Adjuvant Therapy for uLMS

The only effective treatment for uLMS is complete early resection. There is no established adjuvant chemotherapy regimen administered after complete resection [8], and adjuvant radiation therapy is ineffective [9]. Chemotherapy is the main treatment for unresectable, advanced, or recurrent cases.

The standard surgical procedures for resectable cases are total abdominal hysterectomy and bilateral salpingo-oophorectomy. However, oophorectomy and lymph node dissection (LND) can be omitted in uLMS [10]. Hematogenous metastasis to the lungs and liver is likely to occur in uLMS from an early stage. However, the rate of metastasis of uLMS in the ovary is as low as 4%, and ovarian preservation can be considered in younger patients with a significant period left before attaining menopause and with a tumor confined to the uterus. Furthermore, the metastasis rate of uLMS to the retroperitoneal lymph nodes is 6–11%, and in patients whose tumor is limited to the uterus, the metastasis rate is 0–4%. Moreover, if lymph node metastasis has already occurred, hematogenous metastasis to the lungs or liver, or peritoneal dissemination, is also likely to have occurred. Therefore, LND has low diagnostic and/or therapeutic significance, and lymph node biopsy is performed if lymph node enlargement has been confirmed using preoperative imaging [11].

In Japan, postoperative adjuvant chemotherapy is administered to approximately 60% of postinitial operative patients, including patients who underwent total resection. In JGOG 2049s, a retrospective study examining the actual treatment and treatment results of 307 patients with uLMS in Japan, the absence of adjuvant therapy is an independent poor prognostic factor (hazard ratio [HR]: 1.62, 95% confidence interval [CI]: 1.12–2.34, *p* = 0.010). Approximately 40% of patients who underwent adjuvant chemotherapy regimens chose gemcitabine-docetaxel combination therapy (GD therapy) [7]. However, despite several meta-analyses on this matter, a definite conclusion has not been reached on the association between postoperative adjuvant chemotherapy and the prognosis of patients [8]. As such, a standard adjuvant chemotherapy regimen is yet to be established.

### 3.3. First-Line Chemotherapy for Advanced or Recurrent uLMS

No chemotherapy regimen has surpassed doxorubicin monotherapy (Dox therapy) in terms of OS in phase 3 trials as the first-line chemotherapy for unresectable, advanced, or recurrent cases. Table 2 shows the results of major phase 3 trials of first-line chemotherapy for advanced or recurrent STS, including uLMS. The OS of Dox therapy has exceeded 20 months according to recent trials [12,13,14].

To date, although various combinations or molecular-targeting drugs have surpassed Dox therapy in terms of overall response rate (ORR) or progression-free survival (PFS), an improvement in OS is yet to be observed. Phase 3 randomized controlled trials (RCTs) for uLMS alone have not been performed due to its rarity, but the Gynecologic Oncology Group investigated the effect of adding bevacizumab (Bev) to GD therapy for untreated, recurrent, and advanced uLMS [12]. This was because uLMS was more sensitive to GD therapy than other STSs, and the ORR of uLMS was 35.8% in a phase 2 trial [15]. However, the results found no significant difference in terms of OS and PFS, and the accrual was stopped early for futility, which ruled out the additional effect of Bev.

Furthermore, an RCT on Dox therapy and GD therapy for STS including uLMS (GeDDiS trial) did not demonstrate the superiority of GD therapy [13]. In 2019 the results of a randomized phase 3 trial of doxorubicin-oralatumab combination therapy (ANNOUNCE trial), ruled out the additional effect of oralatumab, a monoclonal antibody for a subunit alpha of platelet-derived growth factor receptor, with Dox therapy [14]. Many of these phase 3 trials targeted a wide variety of STSs; however, in a phase 3 trial, an RCT was conducted that compared GD therapy with GD therapy + Bev. Based on the above, Dox therapy is considered the standard first-line chemotherapy regimen for advanced or recurrent uLMS. However, among uLMS patients in the GeDDiS trial, the OSs of patients who underwent Dox therapy and those who underwent GD therapy were considered equal (HR: 1.06, 95% CI, 0.65–1.72, *p* = 0.38).

Alkylating agents such as cyclophosphamide, ifosfamide (IFM), or dacarbazine (DTIC) have been considered to have potential efficacy for certain subtypes of STS, including LMS [11]. Although alkylating agents have been used for LMS in monotherapy or combination therapy with Dox, their effectiveness has not been sufficiently shown in clinical trials [16,17,18], partly because patients with various subtypes are included in phase 3 clinical trials. The European Organization for Research and Treatment of Cancer Soft Tissue and Bone Sarcoma Group conducted a large retrospective study examining the efficacy of IFM or DTIC for Dox-based chemotherapy for advanced LMS. Propensity-score matched analysis revealed that the combination of Dox + DTIC as first-line chemotherapy for advanced LMS was more effective than Dox alone and Dox + IFM; hence, prospective studies are needed to evaluate the efficacy of adding these agents to Dox-based chemotherapy [19].

In addition, the validity of using trabectedin is being investigated as a first-line chemotherapy for advanced and recurrent uLMS patients. Trabectedin is also one of the alkylating agents, a synthetic compound of tetrahydroisoquinoline extracted from the Caribbean encapsulation *Ecteinascidia turbinata*. Takahashi et al. reported that the combined use with Dox enhances the cytotoxic activity against STS cell lines in vitro [20]. In a phase 2 clinical trial of LMS-02, a single-arm trial to evaluate the efficacy of the first-line combination therapy of Dox with trabectedin for uLMS and nuLMS, the ORR of uLMS was 59.6%, and the achieved disease control rate was 87.3% [21]. In the long-term prognostic analysis of the trial, the median PFS and OS in uLMS were 8.3 months (95% CI 7.4–10.3 months), and 27.5 months (95% CI 17.9–38.2 months); while those of nuLMS were 12.9 months (95% CI 9.2–14.1 months) and 38.7 months (95% CI 31.0–52.9 months), respectively [22]. At the ESMO CONGRESS 2021, there was the breaking news of the results of a randomized controlled phase 3 trial examining the superiority of Dox and trabectedin combination therapy over Dox alone; 6 cycles of Dox (75 mg/m^2^) vs. up to 6 cycles of Dox (60 mg/m^2^) with trabectedin (1.1 mg/m^2^) every three weeks followed by maintenance with trabectedin alone for nonprogressive patients. PFS as a primary end point has been significantly improved by the combination of Dox and trabectedin with manageable toxicity in comparison with Dox alone; the median PFS was 13.5 months [95% CI: 11.3–16.7] vs. 7.3 months [95% CI: 6.2–8.3], with adjusted HR: 0.384 [0.27;0.55] and *p* < 0.0001 [23]. A detailed report of this trial is being awaited.

**Table 2 cancers-14-01180-t002:** First-line chemotherapy for advanced or recurrent STS (phase 3 trial, including uLMS).

Regimen	Year	Patients (n)	LMS (n)	uLMS (n)	ORR (%)	mPFS (Months)	mOS (Months)
Dox vs. Dox + CPA [16]	1985	104	38	38	19 vs. 19	5.1 vs. 4.9	11.6 vs. 10.9
Dox + DTIC bolus vs. infusion [17]	1991	240	105	27	17 vs. 17	NA	NA
GD + placebo vs. GD + Bev [12]	2015	107	107	107	31.5 vs. 35.8	6.2 vs. 4.2	26.9 vs. 23.3
Dox vs. GD [13]	2017	257	118	71	19 vs. 20	5.8 vs. 5.9	19.1 vs. 16.8
Dox + placebo vs. Dox + olaratumab [14]	2020	506	234	94	18.3 vs. 14.0	6.8 vs. 5.4	19.7 vs. 20.4

STS, soft tissue sarcoma; uLMS, uterine leiomyosarcoma; n, number; LMS, leiomyosarcoma; ORR, objective response rate; mPFS, median progression-free survival; mOS, median overall survival; NA, not available; Dox, doxorubicin; CPA, cyclophosphamide; DTIC, dacarbazine; GD, gemcitabine + docetaxel; Bev, bevacizumab.

### 3.4. Second-Line Chemotherapy for Advanced or Recurrent uLMS

There are no phase 3 trials on second-line chemotherapy regimens for uLMS alone. At present, drugs covered by health insurance in Japan include pazopanib [24], trabectedin [25], and eribulin [26]. The therapeutic effects of these drugs for uLMS are not sufficient, and the ORR is only 4–11%, PFS is 2.6–4.6 months, and OS is 12.4–13.5 months (Table 3).

Among RCTs in phase 2 on therapies beyond second-line chemotherapy regimens targeting uLMS alone (Table 4), the TAXOGEN trial randomly compared gemcitabine monotherapy (GEM therapy) and GD therapy [29]. The results did not show a significant additive effect of docetaxel with GEM single-agent therapy (GD therapy); the ORR was 19% (compared with 24% for GEM therapy), and the median PFS (4.7 months) did not exceed 5.5 months.

In single-arm, phase 2 clinical trials, GD therapy [30] and an aromatase inhibitor, letrozole [31], were the only treatment regimens that exceeded the expected values of ORR and PFS at 12 weeks, which were the primary end points, respectively. However, the ORR was only 0–27%, and PFS was only 3–6.7 months, highlighting the urgent need for development of novel treatments (Table 4). The therapeutic effect of several molecular-targeted drugs other than letrozole on advanced or recurrent uLMS in the setting of second-line chemotherapy and/or beyond was conducted in phase 2 trials, such as nivolumab (an immune checkpoint inhibitor) [32], alisertib (a selective aurora A kinase inhibitor) [33], and sunitinib (a small-molecule, multi-targeted receptor tyrosine kinase inhibitor) [34], but no results suggestive of efficacy on uLMS were observed (Table 4). In these studies, no biomarker-based patient selection was performed. For example, the tumor mutation burden (TMB) is one of the effective biomarkers for immune checkpoint inhibitors, but the nivolumab trial had no inclusion criteria. A low frequency of TMB-high patients in uLMS was reported [35]. Furthermore, loss of phosphatase and tensin homolog (*PTEN*), which has a frequency of 17.7% in uLMS [36], leads to constitutive activation of the phosphatidylinositol-3-kinase/mammalian target of the rapamycin signaling pathway and to an increase in the expression level of vascular endothelial growth factor (VEGF), resulting in resistance to immune checkpoint inhibitors [37,38]. Thus, the therapeutic effect may be improved when used in combination with an inhibitor of treatment-resistant factor, such as anti-VEGF antibody therapies.

**Table 4 cancers-14-01180-t004:** Secondary chemotherapy for advanced or recurrent uLMS (phase 2 trial).

Regimen	Year	Patients (n)	LMS (n)	uLMS (n)	ORR (%)	mPFS (Months)	mOS (Months)
PTX [39]	2003	48	48	48	8.4	NA	NA
GEM [40]	2004	42	42	42	20.5	NA	NA
GD [30]	2008	48	48	48	27 *	6.7	14.7
GEM vs. GD [29]	2012	90	90	46 **	19 vs. 24	5.5 vs. 4.7	NA
Ixabepilone [41]	2014	23	23	23	0	1.4	NA
Letrozole [31]	2014	27	27	27	0	3 (PFS-12W 50% *)	NA
Nivolumab [32]	2017	12	12	12	0	1.8	NA
Alisertib [33]	2017	23	23	23	0	1.7	14.5
Trabectedin [42]	2018	168	168	168	23.5	4.1 (PFS-6M 35.2% *)	20.6
Thalidomide [43]	2007	29	29	29	0	1.9	8.3
Sunitinib [34]	2009	23	23	23	8.7	1.54	15.1

* Superior to the alternative rate of primary end point in the trial. ** An independent randomized phase 2 trial for uterine leiomyosarcoma and nonuterine leiomyosarcoma. uLMS, uterine leiomyosarcoma; n, number; LMS, leiomyosarcoma; ORR, objective response rate; mPFS, median progression-free survival; mOS, median overall survival; NA, not available; PTX, paclitaxel; GEM, gemcitabine; GD, gemcitabine + docetaxel, PFS-12W, progression-free survival rate at 12 weeks after study entry; PFS-6M, progression-free survival rate at 6 months after study entry.

## 4. Homologous Recombination Deficiency in uLMS

### 4.1. Incidence of Homologous Recombination Deficiency in uLMS

Traditionally, STS is classified into simple (STS 1/3) and complex karyotypes (STS 2/3), and LMS belongs to the latter [44]. Developing molecular-targeted drugs that target simple karyotypes with specific translocations and point mutations is important. However, the development of specific molecular-targeted drugs for LMS with a complex karyotype has not progressed, as sarcoma is a product of genomic instability due to genetic alterations, such as those in tumor protein p53 gene (*TP53*) and retinoblastoma gene (*RB*). Recent developments in gene analysis technology using next-generation sequencing have revealed that uLMS and nuLMS are located close to each other in the phylogenetic tree [45]. However, the gene expression of the DNA repair system increases more frequently in uLMS than in nuLMS, and gene expression of the hypoxia-inducing system increases more frequently in nuLMS [45].

Germline breast cancer susceptibility gene 1/2 (*BRCA1/2*) variation is known to increase the lifetime risk of several cancers, such as breast cancer and ovarian cancer [46], and has been well investigated for its highly ethnic-specific in Ashkenazi Jewish and Polish populations [47]. Recent studies have revealed that ethnic-specific germline *BRCA1/2* variation also exists in other ethnic populations, including Asian populations [48]. However, pan-cancer analysis revealed that uLMS shows characteristic alterations in *BRCA1/2*. The major biallelic inactivation of *BRCA1* and/or *BRCA2* in uLMS, which is one of the most proficient biomarkers for the response of poly(adenosine diphosphate-ribose) polymerase (PARP) inhibitors, is somatic homozygous deletion of *BRCA2*, although germline pathogenic or likely pathogenic mutations, along with loss of heterozygosity (LOH), are major causes of biallelic inactivation of *BRCA1/2* in other gynecologic malignancies, including ovarian cancer [49]. To date, the association between somatic *BRCA1/2* variation in uLMS and ethnic-specificity has not been reported, but the prognosis of uLMS with loss of DNA repair-related genes was worse than that of uLMS without alterations in these genes [50].

Table 5 summarizes the previous studies that investigated the frequency of *BRCA1/2* gene mutation in uLMS. Homologous recombination deficiency (HRD), including the pathogenic mutation in *BRCA2*, was frequently observed, and *BRCA2* deletion was observed in 50% (hemizygous deletion 40%, homozygous deletion 10%) of cases in uLMS [51]. Moreover, mutations in the *BRCA2* gene were observed more frequently in uLMS than in nuLMS (approximately 10% vs. 1%) [50,52]. In a study where a cancer genome profile test was conducted on 80 cases of uLMS, Hensley et al. reported that homozygous deletions and truncating mutations in *BRCA2* were observed in 5% and 2.5% of cases, respectively [32]. The frequency of *BRCA* gene mutations in uLMS reported to date ranges from 7.5% to 10%, and abnormalities in the homologous recombination repair (HRR)-related gene are reported to be approximately 18% overall [50].

In addition, gene alterations related to HRR were observed at a high rate; mutations in HRR-related genes, such as ataxia-telangiectasia mutated gene (*ATM*) (40%) and cyclin-dependent kinase 11 (*CDK11*) (30%), were present in 80% of patients [51]. In an analysis of the HRD score in 214 STS samples from the Cancer Genome Atlas database, which calculated an unweighted sum of LOH, telomeric allelic imbalance, and large-scale state transitions, the optimal cutoff value that correlated with clinical HRD was a HRD score of 34.5, and STS with a high HRD score had a significantly worse prognosis than that with a low HRD score [53]. Thus, HRD may be the most promising therapeutic target for uLMS.

**Table 5 cancers-14-01180-t005:** Frequency of *BRCA* gene mutation in uLMS.

Authors	Patients (n)	Methods	Deletion	Truncating Mutation
Chudasama et al. [51]	10	WES	Total: 5 (50%)	0
			Homozygous: 1 (10%)	
			Hemizygous: 4 (40%)	
Rosenbaum et al. [54]	121	MSK-IMPACT™	Total: 8 (7%)	3 (2.5%)
			Homozygous: 8 (7%)	
Hensley et al. [32]	80	MSK-IMPACT™	Total: 4 (5%)	2 (2.5%)
			Homozygous: 4 (5%)	
Seligson et al. [52]	61	Database *	6 (10%)	0

* The Cancer Genome Atlas data were analyzed: uLMS, uterine leiomyosarcoma; n, number; WES, whole exome sequencing; MSK-IMPACT™, a targeted tumor-sequencing test for integrated mutation profiling of actionable cancer targets developed by the genome scientists, bioinformaticians, and molecular pathologists at Memorial Sloan Kettering Cancer Center.

### 4.2. Effect of PARP Inhibitors on uLMS

For tumors with HRDs, such as *BRCA1/2* mutations, PARP inhibitors are being increasingly studied and approved for treatment. In the event of DNA damage, a single-strand break will be repaired by base excision repair (BER) mechanism. PARP1/2 is one of the enzymes involved in BER, and when its function is inhibited by a PARP inhibitor, repair of a single-strand break is suppressed. The DNA is further damaged, and a double-strand break (DSB) occurs, which is repaired by the HRR mechanism. *ATM* detects DSB and attracts repair proteins, such as *BRCA1/2* to the site. DSB cannot be repaired in cells with deletions or truncating mutations in HRR-related genes, resulting in apoptosis (Figure 2). For gynecologic tumors, olaparib, a PARP inhibitor, is covered by medical insurance in Japan for ovarian cancer with a high incidence of HRD. Olaparib is therapeutically effective as a maintenance therapy based on germline or somatic *BRCA* mutations in the tumor tissue or platinum drug sensitivity [55,56]. Niraparib is also a PARP inhibitor with a higher PARP-trapping potency [57]. In addition to being used as a maintenance therapy or adjuvant therapy for both newly diagnosed ovarian cancers with or without HRD [58] and recurrent ovarian cancers with HRD [59], niraparib exhibited antitumor effects when used as a single agent for recurrent ovarian cancer, with HRD in tumor tissue as an index, independent of platinum-drug sensitivity [60].

In preclinical studies, PARP inhibitors showed antitumor effects not only in ovarian cancers but also on uterine sarcoma cell lines with *BRCA2* gene mutations (SK-UT-1 and SK-UT-1B) in a concentration-dependent manner [51]. This cytotoxic effect was synergistic when PARP inhibitors were used in combination with cisplatin. Furthermore, PARP inhibitors showed a similar cytotoxic effect on uterine sarcoma-derived MES-SA cell lines with *NAMPT* and *PTEN* deletions, despite the absence of abnormalities in the *BRCA2* gene [51]. In addition, in a report examining the efficacy of niraparib on cell lines of STS including uLMS, the concentration-dependent antitumor effect was also observed in vitro on a fibrosarcoma cell line (HT-1080) and a uLMS cell line (SK-LMS-1), which were confirmed to be HRD-positive using increased expression of *RAD51* during drug-induced DNA damage as an indicator [53]. Mutations in *BRCA1* or *BRCA2* have not been reported in HT-1080 and SK-LMS-1. Thus, the antitumor effects of niraparib can be expected on HRD-positive sarcomas even without *BRCA* mutations, including uLMS.

Table 6 summarizes clinical reports on administering PARP inhibitors to patients with uLMS. A case series has reported that PARP inhibitors were clinically useful in four patients with *BRCA2* alterations (three patients with deletion and one patient with truncating mutation). All four cases had a previous history of chemotherapy with at least four lines of treatment, but partial response was noted in one case, whereas stable disease was noted in three cases. Moreover, all cases maintained stable disease for a period of 12 months or longer [52]. Another group also reported the effect of PARP inhibitors in six patients with *BRCA2* alterations (four patients with deletions and two patients with truncating mutations accompanied by LOH). Complete response was noted in one case, and partial response was observed in one case; the treatment period was 6–28 months [32]. Based on the aforementioned preclinical and clinical studies, prolongation of OS can be largely expected with PARP inhibitors for HRD-positive uLMS if the patient’s general condition is stable even if the desired response is not obtained.

### 4.3. Clinical Trials for Unresectable Advanced or Recurrent uLMS

There is no information concerning ongoing phase 3 trials on first-line chemotherapy regimens for uLMS at present. Treatments apart from nonplatinum-based regimens, including GD therapy and Dox therapy, which are considered current first-line chemotherapy regimens, are unlikely to become the standard therapy soon. At present, a phase 3 trial (NCT02997358) comparing Dox therapy and Dox + trabectedin therapy + trabectedin maintenance therapy as first-line therapy for unresectable cases has completed enrollment and entered the follow-up period. In this trial, trabectedin may possibly be used in combination starting from first-line chemotherapy for unresectable cases.

As for trials beyond second-line chemotherapy, an RCT (NCT03016819) of AL3818 (anlotinib), which is a novel tyrosine kinase inhibitor selectively targeting VEGFR-2, VEGFR-3, FGFR-1, FGFR-2, FGFR-3, and FGFR-4, along with DTIC, will be conducted. Phase 2 trials, including the trial of avelumab and GEM combination therapy (NCT03536780), the trial evaluating the additional effect of pazopanib monotherapy (NCT02203760), and the trial targeting solid tumors using HRR genetic alteration and immune cells as biomarkers (NCT03851614), have made progress.

## 5. Designs for Future Clinical Trials

Developing new drug therapies for unresectable advanced or recurrent uLMS is an urgent clinical challenge. Although the therapeutic effects of various antitumor agents and molecular-targeted drugs have been investigated, none have shown a sufficient tumor-reducing effect for patients with a previous history of chemotherapy, especially two or more lines. In uLMS clinical trials conducted to date, biomarker-based patient selection has not been performed, but in the future, clinical trials based on pharmacogenomics are required. Hence, clinical trials based on the estimand of whether the treatment selection based on efficacy biomarker improves the therapeutic effect on uLMS patients are needed. Although some molecular-targeted drugs are expected to maintain a stable disease for a longer period even without tumor shrinkage, setting a time-to-event as the primary end point is difficult because of the rarity of uLMS, as when setting the primary end point to a time-to-event, such as PFS or OS, efficacy must be proved through RCTs in general.

We simulated the number of cases required when planning an RCT with PFS as the primary end point. Using SAS version 9.4 (SAS Institute Inc., Cary, NC, USA), the simulation was performed from PFS in the TAXOGEM trial [29] if PFS follows an exponential distribution. The hazard of the GEM-alone group is λ1 = 0.1475 (median PFS, 4.7 months), and the hazard of the GD group is λ2 = 0.1260 (median PFS, 5.5 months). For each group of exponential random numbers and random numbers of 20–25, we generated 50, 100, 300, 600, and 1000 cases. We repeated this 100 times, accumulated the *p*-values of the log-rank test, and drew a histogram of the *p*-values. The power, however, did not reach 0.8. When calculated again by PASS version 14.0 (NCSS, LLC, Kaysville, UT, USA), the number of cases required to secure the power of 0.8 was 1940 in each group. Similarly, we calculated with PASS version 14.0, using a 2.6-month PFS found in the eribulin trial [26]. The power was 0.8, the significance level was 0.025 on one side, and the numbers of events that occurred were 36 and 21, respectively. The required number of cases was 68 in each group (136 patients). Thus, it may be impossible to set an RCT targeting uLMS patients with a specific biomarker.

Unlike cytotoxic antitumor agents, some molecular-targeted drugs may significantly affect disease stability rather than tumor shrinkage. Glabbeke et al. provided indicators of the PFS rates to be set as primary end points in the single-arm, phase 2 trial for STSs [61]. They reported that a 20% PFS rate at 3 months (PFS-3M) or a 7% PFS rate at 6 months (PFS-6M) after the treatment in second-line chemotherapy indicates an ineffective drug and that 40% and 15% for PFS-3M and PFS-6M are considered active drugs, respectively. Following this report, PFS-3M and PFS-6M have been used as the primary end points of several single-arm, phase 2 clinical trials for uLMS. However, these indicators are based on the RCTs reported before 2000. Several drugs have improved the PFS of STS patients with a previous history of chemotherapy in recent RCTs [24,25,26]; thus, whether using similar indicators in future trials is appropriate is arguable.

Therefore, ORR can still be a helpful indicator as to the primary end point in a single-arm, phase 2 study investigating the efficacy of molecular-targeted therapy for uLMS with specific biomarkers. However, it is necessary to recognize that there is a risk that the effects of drugs that are effective for long-term lesion maintenance cannot be detected. In uLMS, for which RCTs cannot be set due to its rarity, it is necessary to devise a trial design for conducting a single-arm, phase 2 study in which time-to-event is set as the primary end point.

## 6. Conclusions

Considering the lack of trials targeting *BRCA* alteration or positivity of HRD in uLMS, it is important to verify the efficacy of PARP inhibitors for *BRCA* mutation-positive cases or *BRCA* mutation-negative and HRD-positive cases with a previous history of receiving at least one line of chemotherapy for unresectable, advanced, or persistent uLMS, which currently do not have a standard therapy regimen. The JGOG are preparing to conduct a new investigator-initiated trial to investigate the efficacy and safety of niraparib monotherapy for uLMS with HRD. The research presented in the study focuses on recurrent and intractable cases with a poor prognosis and limited effects of existing drug regimens.

## Figures and Tables

**Figure 1 cancers-14-01180-f001:**
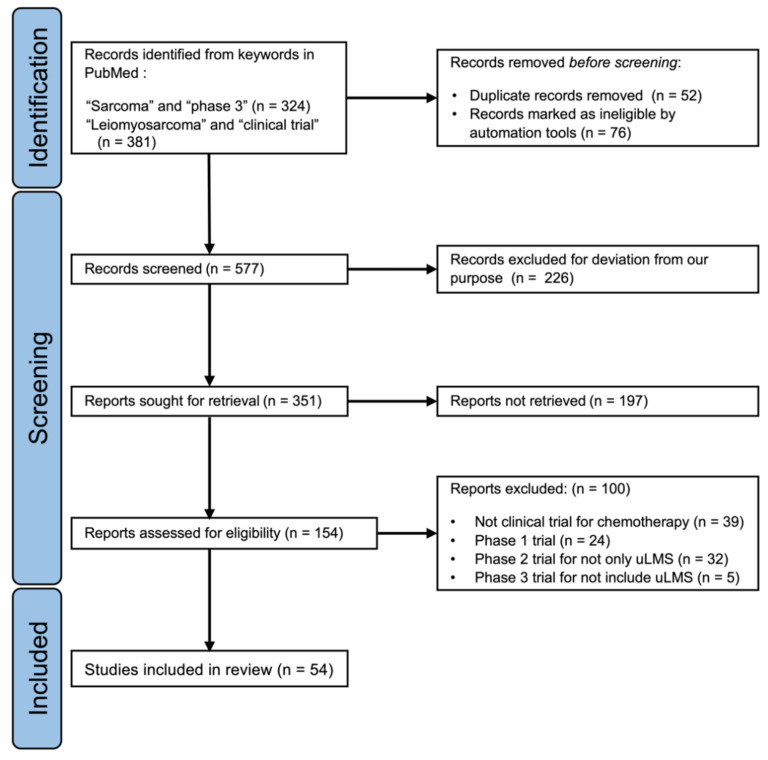
PRISMA 2020 flow diagram for search and identification of clinical trials of soft tissue sarcoma, including uterine leiomyosarcoma, via PubMed database.

**Figure 2 cancers-14-01180-f002:**
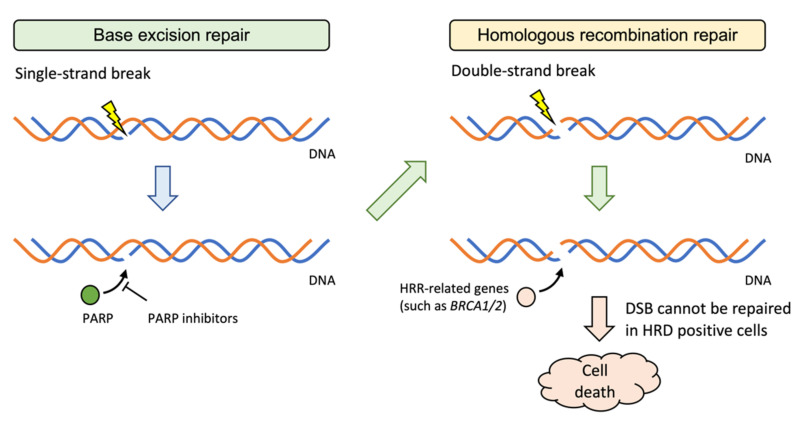
PARP inhibitors for HRD positive cells.

**Table 1 cancers-14-01180-t001:** uLMS staging and overview of prognosis.

FIGO 2008	Primary Tumor	Distribution (%) [6]	5-year OS (%) [7]
Stage I	Tumors limited to the uterus	65.7	57.1
IA	Tumor size ≤ 5 cm		
IB	Tumor size > 5 cm		
Stage II	Tumor extends beyond the uterus, within the pelvis	9.8	44.6
IIA	Tumor involves the adnexa		
IIB	Tumor involves other pelvic tissues		
Stage III	Tumors infiltrates abdominal tissues	7.8	20.4
IIIA	One site		
IIIB	More than one site		
IIIC	Regional lymph node metastasis		
Stage IV		16.7	23.2
IVA	Invasion of bladder and/or rectum		
IVB	Distant metastases		

FIGO, International Federation of Gynecology and Obstetrics.

**Table 3 cancers-14-01180-t003:** Secondary chemotherapy for advanced or recurrent STS (phase 3 trial).

Line	Regimen	Year	Patients (n)	LMS (n)	uLMS (n)	ORR (%)	mPFS (months)	mOS (months)
Third	Pazopanib vs. placebo [24]	2012	369	165	NA	6 vs. 0	4.6 vs. 1.6 *	12.5 vs. 10.7
Second	Ridaforolimus vs. placebo [27]	2013	711	231	NA	(CBR: 40.6 vs. 28.6)	4.4 vs. 3.65 *	22.7 vs. 21.3
Second	Ombrabulin + CDDP vs. placebo + CDDP [28]	2015	355	95	NA	4 vs. 1	1.54 vs. 1.41 *	11.44 vs. 9.33
Second	Trabectedin vs. DTIC [25]	2016	518	378	212	9 vs. 10	4.2 vs. 1.5 *	12.4 vs. 12.9
Third	Eribulin vs. DTIC [26]	2016	452	297	131	4 vs. 5	2.6 vs. 2.6	13.5 vs. 11.5 *

* *p* < 0.05. STS, soft tissue sarcoma; n, number; LMS, leiomyosarcoma; uLMS, uterine leiomyosarcoma; ORR, objective response rate; mPFS, median progression-free survival; mOS, median overall survival; NA, not available; CBR, clinical benefit rate (proportion of patients achieving complete response, partial response, or stable disease for ≥4 months); CDDP, cisplatin; DTIC, dacarbazine.

**Table 6 cancers-14-01180-t006:** Clinical activity of PARP inhibitors on uLMS with *BRCA2* deletions or truncating mutations.

Authors	Year	Patients (n)	Line of Previous Chemotherapy	Response
Seligson et al. [52]	2019	4	Third: 2 (50%)	PR: 1 (25%)
			Fourth: 1 (25%)	SD: 3 (75%)
			Fifth: 1 (25%)	
Hensley et al. [32]	2020	6	NA	CR: 1 (17%) PR: 1 (17%) All had at least some radiographic regression

PARP, poly(adenosine diphosphate-ribose) polymerase; uLMS, uterine leiomyosarcoma; n, number; CR, complete response; PR, partial response; SD, stable disease; NA, not available.

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
