# Peer review of "Status of the Current Treatment Options and Potential Future Targets in Uterine Leiomyosarcoma: A Review"

_cancers, 2022, doi:10.3390/cancers14051180_

Round 1
Reviewer 1 Report
The manuscript titled “Status of uterine leiomyosarcoma treatment: A review” attempted to review the treatment modalities of a rare form of cancer.
As such the novelty of the manuscript is not that high, so authors need to highlight in a subsection how this review is different than the published work. In addition, there are some deficiencies in the manuscript which need to be addressed to make it publishable:
- The abstract doesn’t have the purpose of this manuscript listed
- In general the review lacks depth and critical analyses of the current literature and clinical information in this area.
- A figure and corresponding discussion are needed to depict the characteristics of uterine leiomyosarcoma.
- Page 2, section, 2, a flowchart is needed to describe the literature search strategy. What were the inclusion and exclusion criteria for the articles included in this manuscript?
- Table 5 discusses the gene mutation among patients. What were the ethnic distribution of these mutations? What were the treatment outcomes and toxicity profile in these patients with gene mutation?
- Page 8, Section 5, role of pharmacogenomics in the design of future trials should be discussed.
Reviewer 2 Report
This timely and interesting review by Hiroshi Asano, et al. summarized the current standard therapies for uLMS, including surgical procedures and adjuvant chemotherapy. The authors also discussed the genetic alterations in ulMS tumors and the potential therapeutic strategies that target such mutations. Overall, this review is well written and comprehensive. It is suitable for publications pending clarification and further discussion of the following points:
- The authors cited a paper published in 2006 and stated that “less than 10,000 cases of STS occur annually in the US”. However, according to “cancer statistics, 2021”, the incidence of STS is more than 13,000 in the US. The authors should use a more updated demographics of STS.
- To serve the broader audience, the authors should explain the abbreviations when they first show up, e.g. ATM, CDK11 genes.
- In section 4, the authors discussed the targeted therapy in uLMS, especially the ones that target BRCA mutation, such as PARP inhibitors. To help the reader understand better the relationship between homologous recombination deficiency, BRCA, and PARP inhibitor, the author should include the molecular mechanisms of such terms.
- Although the manuscript reads well overall, it requires close copy editing to correct frequent errors in noun/verb agreement.
Reviewer 3 Report
Nice review. I would suggest two additions:
- Discussing the lack of sensitivity of leiomyosarcomas to immune checkpoint inhibitors and strategies being used in that regard.
- Discussing the controversy surrounding the use of ifosfamide, either as single agent or in combination with doxorubicin.
